# Predictive value of CD86 for the occurrence of sepsis (Sepsis-3) in patients with infection

**Dan lv, Keji Zhang** ID *, **Changqing Zhu, Xinhui Xu, Hao Gong, Li Liu**

Department of Emergency, Renji Hospital, Shanghai Jiao Tong University School of Medcine, Shanghai, China

* zhangkj1977@163.com

**Data Availability Statement:** All relevant data are within the manuscript.

**Funding:** Our work was supported by the Science and Technology Support Project for Medical Guidance(Chinese and Western Medicine) of

## Abstract

This prospective observational study explored the predictive value of CD86 in the early diagnosis of sepsis in the emergency department. The primary endpoint was the factors associated with a diagnosis of sepsis. The secondary endpoint was the factors associated with mortality among patients with sepsis. It enrolled inpatients with infection or high clinical suspicion of infection in the emergency department of a tertiary Hospital between September 2019 and June 2021. The patients were divided into the sepsis and non-sepsis groups according to the Sepsis-3 standard. The non-sepsis group included 56 patients, and the sepsis group included 65 patients (19 of whom ultimately died). The multivariable analysis showed that CD86% (odds ratio [OR] = 1.22, 95% confidence interval [CI]: 1.04–1.44, P = 0.015), platelet count (OR = 0.99, 95%CI: 0.986–0.997, P = 0.001), interleukin-10 (OR = 1.01, 95%CI: 1.004–1.025, P = 0.009), and procalcitonin (OR = 1.17, 95%CI: 1.01–1.37, P = 0.043) were independent risk factors for sepsis, while human leukocyte antigen (HLA%) (OR = 0.96, 05%CI: 0.935–0.995, P = 0.022), respiratory rate (OR = 1.16, 95%CI: 1.03–1.30, P = 0.014), and platelet count (OR = 1.01, 95%CI: 1.002–1.016, P = 0.016) were independent risk factors for death in patients with sepsis. The model for sepsis (CD86%, platelets, interleukin-10, and procalcitonin) and the model for death (HLA%, respiratory rate, and platelets) had an area under the curve (AUC) of 0.870 and 0.843, respectively. CD86% in the first 24 h after admission for acute infection was independently associated with the occurrence of sepsis in the emergency department.

## Introduction

Sepsis is a life-threatening organ dysfunction caused by an imbalance in the host's response to infection [1]. In 2017, 48.9 million patients were hospitalized for sepsis worldwide; of these, 11 million patients reportedly died, accounting for 19.7% of the total global deaths [2]. Early recognition and standardized management of sepsis have been proven to reduce the mortality and length of hospital stay of patients with sepsis [3, 4]. However, because of the heterogeneity of the sources of sepsis, pathogens, and host responses, the clinical manifestations of sepsis can vary widely among patients, making early and accurate diagnosis difficult [5, 6].

Studies on the pathogenesis of sepsis have increasingly focused on new sepsis biomarkers to achieve early diagnosis and successful management of sepsis, but the results are inconsistent

Shanghai Municipal and the Science and Technology Commission in 2018 [No.18411967000]. The role of funder is The funders had no role in study design, data collection and analysis, decision to publish, or preparation of the manuscript.

**Competing interests:** The authors have declared that no competing interests exist.

and sometimes conflicting [7–9]. Because sepsis is caused by a disordered host response to infection, biomarkers related to immune dysfunction have received attention in the study of sepsis in recent years [10]. In 2019, the authors established the STAPLAg scoring system, which can be used to predict the occurrence of sepsis and subsequent death using six clinical indicators related to the host's infection and immune status (sodium concentration, troponin I concentration, albumin [ALB] concentration, platelet count [PTL]/lymphocyte ratio, lactate concentration, and age) [11]. Still, the model is not perfect and can be improved.

CD86/CD80 is an important costimulatory molecule on monocytes. It belongs to the B7 family of costimulatory molecules and interacts with CD28 and CD152/cytotoxic T-lymphocyte-associated antigen 4 (CTLA-4) molecules on T cells to affect the survival and response of T cells through co-stimulation and co-inhibition pathways involved in many immune-related diseases [12, 13]. A study by the author's group showed that CD86 gene polymorphisms are related to the expression of CD86 in peripheral blood monocytes and can increase the susceptibility of the human body to sepsis caused by pneumonia [14]. The potential value of CD86 for sepsis prediction has not been explored so far.

The human leukocyte antigen (HLA) is another reliable parameter of innate immune function [15–17]. In sepsis, hyporesponsiveness to lipopolysaccharides (LPS) correlates with decreased HLA expression [18, 19], indicating that HLA expression levels might be involved in the pathogenesis, progression, and prognosis of sepsis. Patients with sepsis showing delayed or no improvement or, even worse, a decline in monocyte HLA expression have a higher risk of adverse outcomes [20].

Therefore, this study aimed to explore the value of the costimulatory molecule CD86 in the early diagnosis of sepsis in the emergency department and the value of HLA in the prognosis of patients with sepsis. The primary study endpoint was the factors associated with a diagnosis of sepsis. The secondary study endpoint was the factors associated with mortality among patients with sepsis.

## Material and methods

### Study design and patients

This prospective observational study enrolled inpatients with infection or high clinical suspicion of infection in the emergency department of Renji Hospital Affiliated to Shanghai Jiaotong University School of Medicine, between September 2019 and June 2021. The ethics committee of Renji Hospital approved the study (2018–210). All patients (or their legal representatives) provided written informed consent for this study.

The inclusion criteria were 1) ≥18 years of age and 2) confirmed infection or high suspicion of infection with clear clinical symptoms. Patients who were highly suspected of having an infection with clear clinical symptoms were also required to meet at least two of the following four diagnostic criteria for systemic inflammatory response syndrome [21]: 1) body temperature of >38°C or <36°C, 2) heart rate of >90 beats/min, 3) respiratory rate of >20 breaths/min, or partial pressure of carbon dioxide of <32 mmHg, and 4) peripheral white blood cell count of >12,000/mm$^3$ or <4000/mm$^3$.

The exclusion criteria were 1) systemic immune disease, 2) advanced malignant tumor, 3) human immunodeficiency virus infection, 4) use of hormones higher than the replacement dose, or 5) immunosuppressive therapy for >1 month.

All patients with infection (or high suspicion of infection) were divided into sepsis and non-sepsis (control) groups according to the Sepsis-3 standard [13]. Patients not meeting the sepsis criteria at admission but developing sepsis during hospitalization were grouped in the sepsis group.

All patients diagnosed with sepsis and septic shock were managed in accordance with the Surviving Sepsis Campaign guidelines, including a 3-h bundle treatment, timely identification and control of the sources of infection, timely retention of culture specimens, rational use of antibiotics and glucocorticoids in the early stage, blood glucose control, respiratory support, organ protection, and prevention and treatment of the complications [22]. Ultimately, the patients with sepsis were divided according to their final outcome: survival or death.

### Laboratory examination

Within 24 h after admission, all patients underwent venous blood draws to measure C-reactive protein (CRP), PTL, procalcitonin (PCT), ALB, lactic acid, white blood cells, lymphocytes, monocytes, interleukin (IL)-2, IL-6, IL-10, IL-17A, and soluble IL-2 receptor (sIL-2R). These biomarkers were measured after admission and before the first dose of antibiotics.

Whole blood samples were collected within 24 h after admission. The blood cells and plasma were separated within 1 h. Then, 20% dimethyl sulfoxide cryoprotectant was added, and the samples were stored at -80°C. The expression levels of CD86 and the immunosuppressive index HLA subtype DR (HLA-DR) in the peripheral blood mononuclear cells (PBMCs) of all patients were detected by flow cytometry and recorded as CD86% and HLA-DR%. A reverse transcription-polymerase chain reaction was used to detect the CD86 mRNA levels in PBMCs.

### Data collection

Patients' clinical data were recorded, including age, sex, comorbidities, site of infection, etiological examination results, Acute Physiology and Chronic Health Evaluation II (APACHE II) score, and Sequential Organ Failure Assessment (SOFA) score at admission [1]. The CRP/ALB ratio, neutrophil/lymphocyte ratio (NLR), platelet/lymphocyte ratio, and lymphocyte/monocyte ratio (LMR) were calculated. The highest SOFA score during hospitalization and the macrohemodynamic parameters of the patients in the sepsis group were recorded. The admission and duration of stay in the intensive care unit (ICU) and the survival state were also recorded.

### Statistical analysis

Statistical analysis was performed using SAS 9.4 (SAS Institute, Cary, NC, USA). A normality test was conducted for the continuous data. The data that met the criteria for normal distribution were *presented* as means ± standard deviations (SD) and compared using Student's t-test. The continuous that failed to meet the criteria for normality were presented as medians (Q1-Q3) and compared using the Wilcoxon test. The categorical data were presented as n (%) and compared using the chi-square test or Fisher's exact test, as appropriate. Logistics regression models were used for the multivariable analyses. Variables with $P < 0.15$ in the univariable regression analyses were selected for the multivariable logistics regression analyses. The predictive values of CD86 and HLA were evaluated using the area under the receiver operating characteristics (ROC) curve (AUC). A two-tailed $P < 0.05$ was considered statistically significant.

### Results

Of the 130 enrolled patients, nine were excluded (Fig 1). The non-sepsis group included 56 patients, and the sepsis group included 65 patients (19 of whom ultimately died). Twenty-

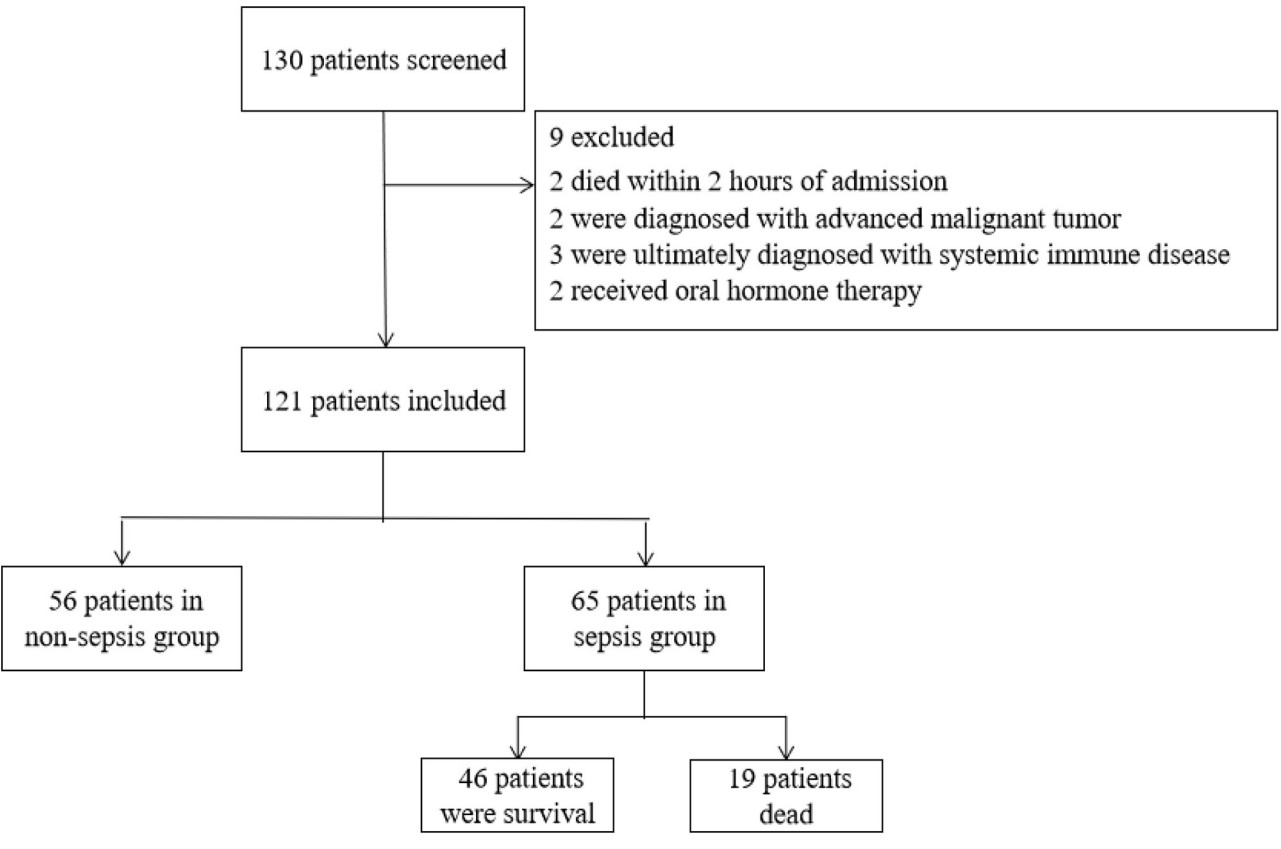

**Fig 1. Study flowchart.**

three patients changed from non-sepsis to sepsis during hospitalization. They were grouped in the sepsis group.

There were no statistically significant differences in age, sex, or body mass index among the groups (all P>0.05), but the proportion of men was high in all groups (67.4% of the patients with sepsis and survived, 68.4% of the patients with sepsis who died, and 62.5% of the patients without sepsis). The main primary infection site in the sepsis and non-sepsis groups was the respiratory system, followed by the abdomen (intestines, liver and gallbladder, and urinary system). Twelve patients had secondary bloodstream infections (10 in the sepsis group and two in the non-sepsis group). Patients with sepsis were likelier to develop shock or require ventilator-assisted ventilation (Table 1).

The CD86%, neutrophil count, CRP/ALB, NLR, IL-6, IL-10, sIL-2R, CRP, and PCT levels were significantly higher, and the HLA-DR%, lymphocytes, PLT, and ALB levels were significantly lower in the sepsis group than in the non-sepsis group (all P<0.05). In the sepsis group, the CD86% and NLR were significantly higher, and the HLA-DR% and LMR were significantly lower in the patients who died than in those who survived (Fig 2 and Table 2).

The multivariable analysis showed that CD86% (odds ratio [OR] = 1.22, 95% confidence interval [CI]: 1.04–1.44, P = 0.015), PLT (OR = 0.99, 95%CI: 0.986–0.997, P = 0.001), IL-10 (OR = 1.01, 95%CI: 1.004–1.025, P = 0.009), and PCT (OR = 1.17, 95%CI: 1.01–1.37, P = 0.043) were independent risk factors for sepsis (Table 3). In the patients with sepsis, the multivariable logistics regression showed that HLA% (OR = 0.96, 05%CI: 0.935–0.995, P = 0.022), respiratory rate (OR = 1.16, 95%CI: 1.03–1.30, P = 0.014), and PLT (OR = 1.01,

**Table 1. Characteristics of the patients (n = 121).**

| Characteristics | Sepsis (n = 65) | | | Non-sepsis (n = 56) | P |
|---|---|---|---|---|---|
| | Survive (n = 46) | Death (n = 19) | Total (n = 65) | | |
| Age (years) | 65.4±18.2 | 66.0±19.4 | 67.00±16.4 | 65.5 (55.0–81.0) | 0.946 |
| Sex | | | | | |
| Male | 31 (67.4) | 13 (68.4) | 44 (67.7%) | 35 (62.5) | 0.550 |
| Female | 15 (32.6) | 6 (31.6) | 21 (32.3%) | 21 (37.5) | |
| BMI (kg/m$^2$) | 23.9 (22.0–26.7) | 22.2 (17.8–26.0) | 23.78 (21.60–26.12) | 22.7 (20.6–25.8) | 0.384 |
| Infective lesion | | | | | 0.118 |
| Lung | 30 | 16 | 46 | 41 | |
| Abdominal | 23 | 5 | 28 | 18 | |
| Blood | 4 | 6 | 10 | 2 | |
| Central nervous system | 1 | 1 | 2 | 0 | |
| Skin and soft tissue | 1 | 0 | 1 | 2 | |
| Others | 0 | 0 | 0 | 1 (16.67) | |
| Sepsis shock | 10 (35.71) | 18 (64.29) | 28 (43.08) | 0 | <0.001 |
| Mechanical ventilation | 11 (45.83) | 13 (54.17) | 24 (36.92) | 0 | <0.001 |
| CRRT | 1 (20.00) | 4 (80.00) | 5 (7.69) | 1 (16.67) | 0.215 |
| Hospital stay (days) | 13.00 (9.00–19.00) | 15.00 (7.00–24.00) | 13.00 (9.00–20.00) | 10.00 (8.00–15.00) | 0.103 |
| Admission SOFA scores | 3.00 (1.00–7.00) | 7.00 (4.00–15.00) | 4.00 (2.00–7.00) | 0.00 (0.00–1.00) | <0.001 |
| Max SOFA scores | 4.00 (2.00–7.00) | 16.00 (11.00–21.00) | 6 (2.00–9.00) | 0.00 (0.00–1.00) | <0.001 |
| APACHEII scores | 13.70±6.87 | 17.37±7.60 | 14.00±7.42 | 8.50 (6.00–12.50) | <0.001 |

Data are presented as mean ± standard deviation, n (%), or median (Q1-Q3).

BMI, body mass index; CRRT, continuous renal replacement therapy; SOFA, Sequential Organ Failure Assessment; APACHE II, Acute

Physiology and Chronic Health Evaluation II; Max SOFA score: highest SOFA score during emergency hospitalization.

P, sepsis vs. non-sepsis. Note: P<0.05 is statistically significant.

95%CI: 1.002–1.016, P = 0.016) were independent risk factors for death in patients with sepsis (Table 3).

The AUC of the model composed of CD86%, PLT, IL-10, and PCT to predict the occurrence of sepsis was 0.870 (0.800 sensitivity and 0.857 specificity). The AUC of CD86% for the prediction of the occurrence of sepsis was 0.720 (0.754 sensitivity and 0.625 specificity), and the cutoff value was 0.92. In patients with sepsis, the AUC of the model composed of HLA, PLT, and respiratory rate for predicting the death of patients with sepsis was 0.843 (0.684 sensitivity and 0.891 specificity). When HLA% was used to predict death due to sepsis, the AUC was 0.680 (0.998 sensitivity and 0.548 specificity), and the cutoff value was 55.42 (Fig 3).

## Discussion

This study showed that CD86%, PLT, IL-10, and PCT were independent risk factors for sepsis in patients with infection, while HLA%, PLT, and respiratory rate were independent risk factors for death in patients with sepsis. The results suggest that CD86% and HLA-DR% in the first 24 h after admission for acute infection were associated with the occurrence of sepsis and death from sepsis.

In the present study, most cases of sepsis were caused by respiratory infections, consistent with the data on the global incidence of sepsis from 1990 to 2017 reported by Rudd et al. [2]. Indeed, sepsis is more common in children and older adults, and the main sources of infection are the respiratory tract and gastrointestinal tract. The mortality rate of 29% is also consistent with the literature [2, 23, 24].

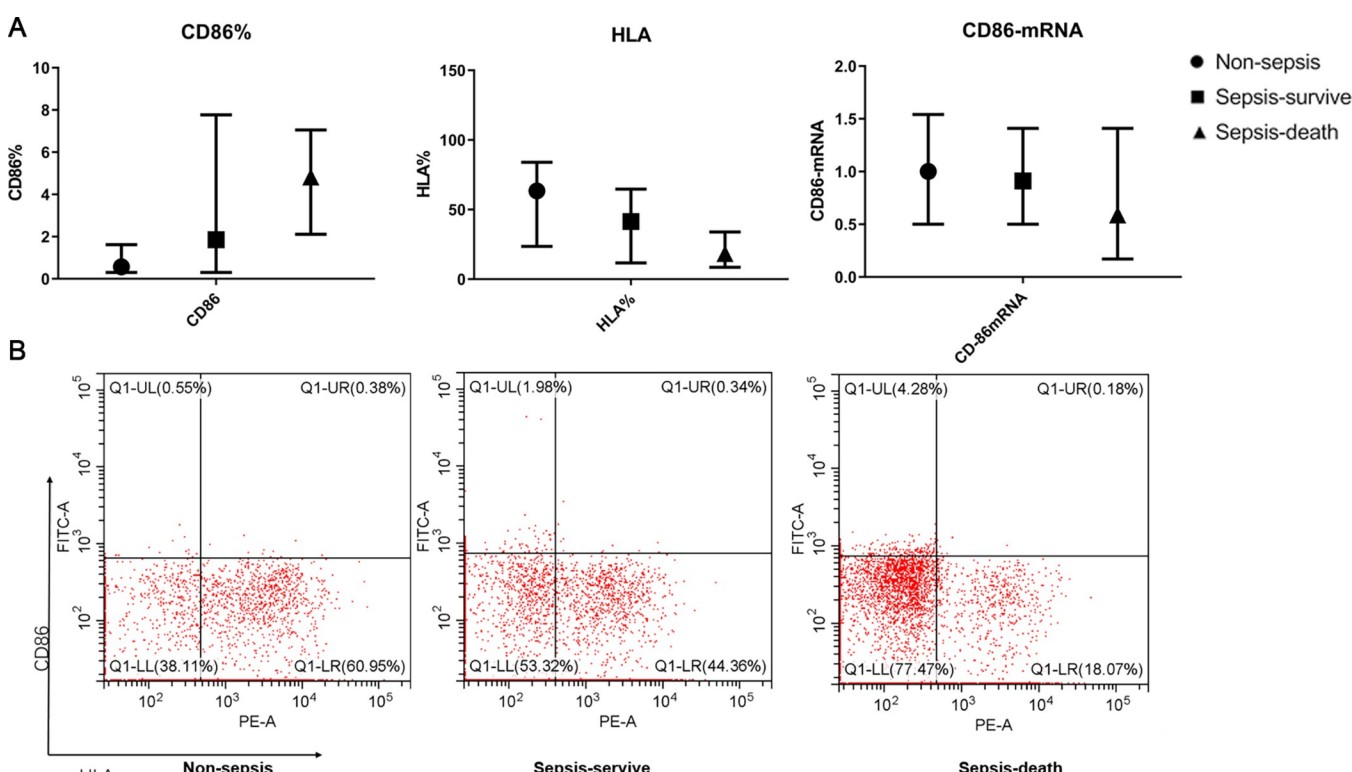

**Fig 2.** (A) CD86%, human leukocyte antigen (HLA%), and CD86 mRNA in the sepsis and non-sepsis groups. CD86% expression was lower in the non-sepsis group and higher in the sepsis-death group. In contrast, HLA% was lower in the sepsis-death group and higher in the non-sepsis group. There were no significant differences in CD86 mRNA among the three groups. (B) Flow cytometry outputs of CD86 among the three groups.

During the development of sepsis, abnormal immune responses affect the production and function of the innate and adaptive immune cells, leading to immune system imbalance. The present study showed significant differences in CD86 expression, neutrophil count, CRP/ALB, NLR, IL-6, IL-10, sIL-2R, CRP, PCT, HLA-DR, lymphocytes, PLT, ALB, and LMR between patients with sepsis and those with non-septic infections, consistent with the presence of immune disorders in patients with sepsis [10].

The above-mentioned immune-related biomarkers can reflect the host's sepsis-induced inflammatory response and immunosuppression. Decreased lymphocytes, decreased expression of monocyte HLA-DR, and overexpression of IL-10 are indicators of sepsis-induced immunosuppression [25–27]. A more valuable indicator for evaluating the risk of immunosuppression and death, namely the NLR, was recently identified in several studies. Indeed, in an observational cohort of 5056 unselected ICU patients, the mortality rate gradually increased as the NLR quartile increased [28]. It was also confirmed in patients with sepsis and severe COVID-19 [29, 30]. The parameters of the predicted score established in the authors' study in 2020 overlap with these indicators [11].

HLA-DR is the major histocompatibility class II molecule with the highest expression level and is involved in antigen presentation [15]. The downregulation of HLA-DR is the most effective marker of sepsis-induced immunosuppression [15, 27]. In 2009, a prospective observational study by Lukaszewicz et al. [31] showed that the HLA-DR was decreased to a greater degree in the sepsis group than in the non-sepsis control group and that this early downregulation of HLA-DR was associated with 28-day mortality. In addition, other studies of patients with burns and trauma showed that the HLA-DR <30% or <5000 antibodies bound per cell

**Table 2. Comparison of peripheral blood indices in sepsis and non-sepsis groups.**

| Characteristics | Non-sepsis (n = 56) | Sepsis (n = 65) | | | P# | P* |
|---|---|---|---|---|---|---|
| | | Survivor (n = 46) | Death (n = 19) | Total (n = 65) | | |
| WBC ($10^9$/L) | 10.26 (6.92–13.21) | 11.57 (8.90–17.51) | 12.33 (9.67–21.79) | 11.87 (9.08–17.51) | 0.035 | 0.177 |
| LY ($10^9$/L) | 1.12 (0.73–1.49) | 0.73 (0.57–1.27) | 0.81 (0.50–0.99) | 0.73 (0.57–1.10) | 0.003 | 0.249 |
| PLT ($10^9$/L) | 222.50 (178.50–316.00) | 142.00 (91.00–177.00) | 203.00 (97.00–296.00) | 148.00 (94.00–218.00) | <0.001 | 0.210 |
| N ($10^9$/L) | 8.84 (5.18–11.12) | 10.09 (6.71–15.46) | 11.07 (6.94–18.88) | 10.23 (6.94–15.46) | 0.017 | 0.253 |
| MONO ($10^9$/L) | 0.67 (0.52–0.96) | 0.71±0.37 | 0.84±0.58 | 0.64 (0.44–1.04) | 0.543 | 0.498 |
| ALB (g/L) | 31.00 (28.05–36.05) | 28.00 (25.30–33.00) | 27.00 (23.50–31.30) | 27.00 (25.00–31.80) | 0.002 | 0.435 |
| Lac (mmol/L) | 1.85 (1.20–2.01) | 1.55 (1.20–2.40) | 2.50 (0.96–2.80) | 1.70 (1.11–2.70) | 0.702 | 0.204 |
| CRP/ALB | 1.33 (0.78–3.63) | 3.65 (1.42–5.85) | 4.32 (1.99–6.98) | 3.65 (1.45–5.99) | <0.001 | 0.419 |
| NLR | 7.51 (3.67–12.90) | 14.03 (6.38–21.56) | 11.41 (7.24–34.17) | 13.45 (7.24–23.24) | 0.001 | 0.371 |
| PLR | 221.21 (149.78–324.30) | 169.87 (100.46–276.74) | 267.71 (100.00–371.01) | 179.37 (100.46–300.00) | 0.067 | 0.248 |
| LMR | 1.47 (0.96–2.32) | 1.40 (0.96–1.82) | 1.04 (0.81–1.75) | 1.32 (0.92–1.81) | 0.177 | 0.039 |
| IL-2 (pg/ml) | 1.78 (1.03–3.60) | 2.31 (1.39–5.16) | 1.67 (1.14–5.16) | 2.06 (1.29–3.60) | 0.174 | 0.643 |
| IL-8 (pg/ml) | 46.90 (22.40–94.14) | 65.93 (27.70–116.24) | 51.50 (26.90–157.00) | 63.64 (27.70–94.14) | 0.249 | 0.679 |
| TNF-α (pg/ml) | 7.34 (1.39–10.08) | 8.28 (2.04–12.98) | 12.98 (1.12–27.30) | 8.47 (1.76–15.40) | 0.152 | 0.495 |
| IFN-γ (pg/ml) | 3.41 (1.92–27.81) | 3.14 (1.68–3.14) | 2.54 (1.37–3.14) | 3.24 (1.68–27.81) | 0.624 | 0.237 |
| IL-17A (pg/ml) | 6.02 (1.85–6.49) | 5.83 (1.93–6.34) | 3.89 (1.33–5.83) | 6.34 (1.64–6.49) | 0.975 | 0.044 |
| IL-6 (pg/ml) | 20.47 (8.97–58.35) | 37.50 (17.00–95.20) | 72.00 (16.70–158.70) | 54.40 (17.00–81.06) | 0.005 | 0.286 |
| IL-10 (pg/ml) | 5.00 (5.00–9.73) | 8.16 (5.00–19.60) | 9.91 (5.00–38.10) | 8.50 (5.00–24.30) | 0.007 | 0.306 |
| SIL-2R (u/ml) | 992.00 (552.00–1806.7) | 1671.0 (902.00–2515.0) | 1716.6 (1171.0–2508.0) | 1709.0 (1042.0–2508.0) | 0.001 | 0.750 |
| CRP (mg/L) | 54.47 (28.68–99.70) | 106.48±81.17 | 114.77±76.89 | 96.01 (43.62–159.13) | 0.005 | 0.589 |
| PCT (ng/ml) | 0.18 (0.10–0.69) | 2.34 (0.20–15.20) | 2.62 (0.37–11.68) | 2.39 (0.28–8.74) | <0.001 | 0.879 |
| CD86% | 0.57 (0.30–1.62) | 1.85 (0.30–7.77) | 4.83 (2.11–7.05) | 2.42 (0.92–7.52) | <0.001 | 0.062 |
| HLA% | 63.46 (23.52–83.89) | 41.38 (11.72–64.67) | 18.47 (8.54–33.99) | 29.61 (11.72–55.27) | <0.001 | 0.002 |
| CD86 mRNA | 1.00 (0.50–1.54) | 0.91 (0.56–1.41) | 0.59 (0.17–1.41) | 0.89 (0.33–1.44) | 0.295 | 0.917 |

Data are presented as median (Q1-Q3).

WBC, white blood cell count; LY, lymphocyte count; PLT, platelet count; N, neutrophil count; MONO, monocyte count; ALB, albumin; Lac, lactic acid; CRP, C-reactive protein; NLR, neutrophil/lymphocyte ratio; PLR, platelet/lymphocyte ratio; LMR, lymphocyte/monocyte ratio; IL, interleukin; TNF-α, tumor necrosis factor-α; IFN-γ, interferon-γ; SIL-2R, soluble interleukin-2 receptor; PCT, procalcitonin; HLA, human leukocyte antigen.

P#, sepsis vs. non-sepsis; P*, survivors vs. death in sepsis group. Note: P<0.05 is statistically significant.

was related to sepsis-related complications [19, 32]. In a study of 209 patients with septic shock, patients with HLA-DR levels of ≤40% could be predicted to develop nosocomial infections with a sensitivity of 60% and specificity of 69% [33]. Other cohort studies confirmed that persistently low HLA-DR levels were associated with adverse outcomes and that HLA-DR in sepsis survivors usually returned to a normal level [34–36]. In the present study, flow cytometry showed that the HLA-DR% was significantly lower in the sepsis group than in the non-sepsis group, and the HLA-DR% decrease degree was more significant in non-survivors. HLA-DR was an independent predictor of 28-day mortality in patients with sepsis, and the AUC was 0.680 with a corresponding cutoff value of 55.42. This finding confirms that HLA-DR is an important indicator of sepsis-induced immunosuppression and a predictor of mortality.

CD86/CD80 is an important costimulatory molecule on monocytes [37]. Wolk et al. [12] showed that the expression level of CD86 was positively correlated with HLA-DR, and ICU patients whose CD86 levels continued to decline were more prone to die of multiple organ failure accompanied by infection. Manjuck et al. [38], using the definition of sepsis developed by

**Table 3. Prediction of occurrence and prognosis of sepsis by multivariable logistic regression.**

| Variables | OR (95%CI) | P |
|---|---|---|
| **Sepsis** | | |
| CD86% | 1.221 (1.040–1.435) | 0.015 |
| HLA% | 0.997 (0.981–1.014) | 0.741 |
| CD86 mRNA | 0.904 (0.729–1.122) | 0.361 |
| PLT | 0.991 (0.986–0.997) | 0.001 |
| IL-10 | 1.014 (1.004–1.025) | 0.009 |
| PCT | 1.171 (1.005–1.365) | 0.043 |
| Intercept | 0.245 | |
| **Death** | | |
| HLA% | 0.964 (0.935–0.995) | 0.022 |
| RR | 1.155 (1.030–1.295) | 0.014 |
| PLT | 1.009 (1.002–1.016) | 0.016 |
| Intercept | 0.007 | |

PLT, platelet count; IL, interleukin; PCT, procalcitonin; HLA, human leukocyte antigen; RR, respiratory rate; S.E., standard error; OR, odds ratio; CI, confidence interval.

Note: P<0.05 is statistically significant.

the American College of Chest Physicians/Society of Critical Care Medicine Consensus Conference in 1991, found that the PBMC CD86 levels were significantly lower in ICU patients with sepsis than in patients without sepsis and healthy people [38]. In the present study, the expression of HLA-DR in patients with sepsis was decreased, consistent with the previous

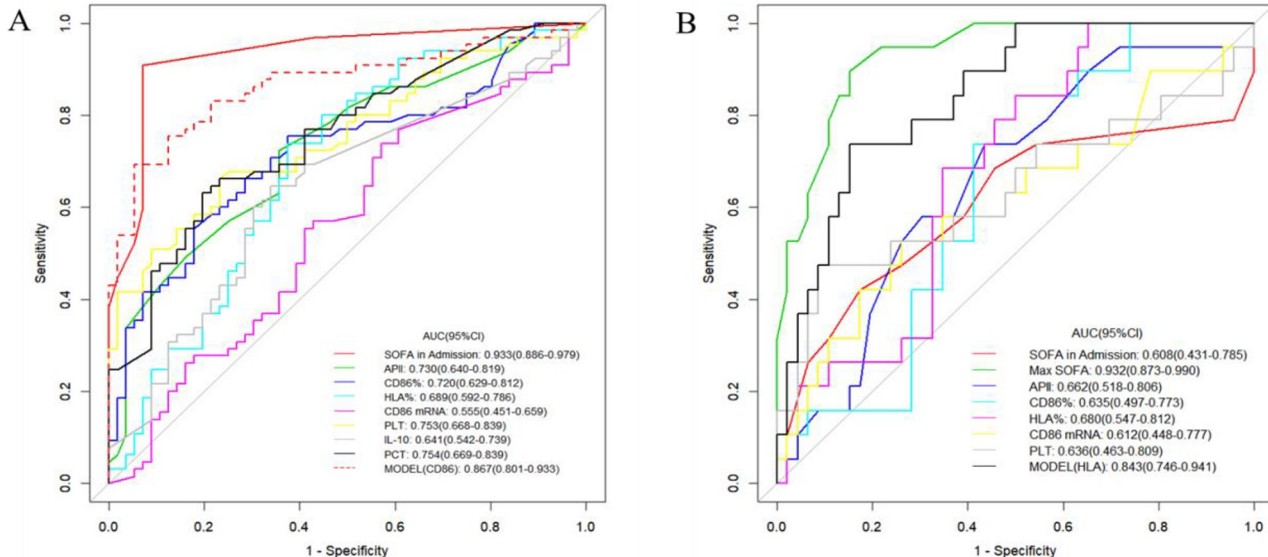

**Fig 3. Area under the receiver operating characteristics (ROC) curve.** There were no statistically significant differences between model CD86 and the SOFA score (used to diagnose sepsis) in predicting the occurrence of sepsis in patients with infection in the emergency department. Max SOFA could effectively predict the occurrence of death in patients with sepsis in the emergency department, but the HLA model was also effective (AUC of 0.843). CD86: cluster of differentiation 86; AUC, area under the curve; CI, confidence interval; APII, Acute Physiology, and Chronic Health Evaluation II score; SOFA, Admission, Sequential Organ Failure Assessment score at the time of admission; Max SOFA, highest SOFA score during emergency hospitalization; PLT, platelet count; PCT, procalcitonin; MODEL (CD86), model composed of CD86%, PLT, IL-10, and PCT; MODEL (HLA), model composed of HLA, PLT, and RR.

studies, but the expression of CD86 in PBMCs was higher in the sepsis group than in the non-sepsis group, especially in non-surviving patients, which is inconsistent with some previous studies [12, 38, 39]. CD86 can affect the survival and response of T cells through co-stimulation and co-inhibition pathways [27, 40]. The T-cell receptor complex recognizes the antigens on the APCs [41]. CD86 plays an important role in the regulation of both positive and negative functions of T cells [37]. It activates T cells through co-stimulation in the early stage of the inflammatory response, leading to proinflammatory factor secretion (e.g., IL-2) and enhanced body's ability to resist infection [42]. Therefore, the early elevation of CD86 in patients with sepsis is consistent with the known immune mechanisms. In this study, CD86 was an independent predictor of the occurrence of sepsis, with an AUC of 0.720 and a corresponding cutoff value of 0.92. Schutz et al. [43] observed a high recurrence rate in patients who had chronic myeloid leukemia with high expression of CD86 in dendritic cells. Takacs et al. [44] found that CD86 positivity indicated a poor prognosis in patients with chronic lymphocytic leukemia, consistent with the present study. In addition, this study was based on the Sepsis-3 diagnostic criteria, and the conclusion is more in line with the pathophysiological characteristics of sepsis and clinical reality.

This study also showed no significant differences in the CD86 mRNA levels in PBMCs between the sepsis and non-sepsis groups. The CD86 mRNA levels of patients who died of sepsis were decreased, but the lack of statistically significant differences among the groups might be related to polymorphisms in the CD86 gene or different post-transcriptional or post-translational regulation. CD86 gene polymorphisms play different roles in the pathogenesis of sepsis, some of which interfere with or increase the expression of CD86 genes in monocytes [14, 45].

This study has some limitations. First, it was a single-center prospective observational study with a small sample size. Second, the CD86% and other data were measured at a single point in time. Their dynamic changes were not monitored. The inflammatory reaction of the host fluctuates over time. The sample size should be expanded in the future, and the dynamic serial measures of CD86 and related indicators should be monitored. In addition, the CD86% in PBMCs was higher in the sepsis group than in the non-sepsis group, which contradicts the results of some previous studies, but this finding is more in line with the pathophysiological characteristics of sepsis. Considering that the mechanism of CD86 in sepsis is still unclear, further multicenter trials and clinical data are required to verify the results. Finally, the variable selection for the multivariable models was based on a P-value threshold of 0.15 in the univariable analyses. Future multicenter studies with a larger sample size are necessary for confirmation.

## Conclusion

CD86 is an independent predictor of the occurrence of sepsis in patients with infection in the emergency department and has a certain value in the early clinical diagnosis of sepsis in such patients. In addition, HLA-DR is an independent predictor of death from sepsis and can be used for the early prediction of the prognosis of patients with sepsis in the emergency department.

## Supporting information

**S1 Checklist. STROBE statement—checklist of items that should be included in reports of observational studies.**
(DOCX)

**S2 Checklist.** *PLOS ONE* **clinical studies checklist.**
(DOCX)

## Author Contributions

**Conceptualization:** Dan lv, Keji Zhang, Xinhui Xu, Li Liu.

**Data curation:** Dan lv, Keji Zhang, Xinhui Xu, Li Liu.

**Formal analysis:** Changqing Zhu.

**Validation:** Hao Gong.

**Visualization:** Hao Gong.

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
