## [Decision Letter · Decision Letter 0]

16 Aug 2023

PONE-D-23-11681Predictive value of CD86 for the occurrence of sepsis (Sepsis-3) in patients with infectionPLOS ONE

Dear Dr. Zhang,

Thank you for submitting your manuscript to PLOS ONE. After careful consideration, we feel that it has merit but does not fully meet PLOS ONE’s publication criteria as it currently stands. Therefore, we invite you to submit a revised version of the manuscript that addresses the points raised during the review process.

We look forward to receiving your revised manuscript.

Kind regards,

Mian Pan

Academic Editor

PLOS ONE

Journal Requirements:

This work was supported by the Science and Technology Support Project for Medical Guidance (Chinese and Western Medicine) of Shanghai Municipal and the Science and Technology Commission in 2018 [No.18411967000]. The sponser had no role in study design, data collection and analysis, decision to publish, or preparation of the manuscript.

Additional Editor Comments:

We have received the reports from our advisors on your manuscript, "Predictive value of CD86 for the occurrence of sepsis (Sepsis-3) in patients with infection", which you submitted to PLOS ONE.

Based on the advice received, the Editors feel that your manuscript could be reconsidered for publication should you be prepared to incorporate major revisions. When preparing your revised manuscript, you are asked to carefully consider the reviewer comments which are attached, and submit a list of responses to the comments.

COMMENTS FOR THE AUTHOR:

Reviewer #1: This study wants to explore the value of the costimulatory molecule CD86 in the early diagnosis of sepsis in the emergency department and explore the value of HLA in the prognosis of patients with sepsis. The authors pointed out that CD86 is an independent predictor of the occurrence of sepsis in patients with infection in the emergency department and has a certain value in the early clinical diagnosis of sepsis in such patients.

The Manuscript needs some minor revisions:

-Improve the English form.

-It is unclear how control patients were enrolled. Do they always come from the emergency room? are they patients with fever? The authors should clarify the description of this group.

- the tables and figures are clear except for figure 2, the symbols are not clear and are small, also the figure should be divided into A and B.

Reviewer #2:

The authors have written a nice manuscript which is a prospective study.

1. Variables with P<0.15 in the univariable regression analysis were selected for multivariable logistic regression analysis (MVA). Why not only variables with significant value p<0.05 were taken for MVA

2. Why were blood samples collected after 1st dose of antibiotics, and not at baseline?

3. How can variables with such low odds ratios [PLT (OR=0.99, 95%CI: 0.986-0.997, P=0.001), IL-10 (OR=1.01)] be statistically significant? What is the outcome of odds ratio for these variables?

4. In Methods - Please write clearly what was the primary end-point of the study. Writing to explore the value of CD86 and HLA-DR in sepsis, is very vague

5.How many patients who were initially classified as "non-sepsis" later turned into "sepsis" during the course of hospital stay? How were these patients accounted for?

This letter contains confidential information, is for your own use, and should not be forwarded to third parties.

Reviewers' comments:

Reviewer's Responses to Questions

**Comments to the Author**

1. Is the manuscript technically sound, and do the data support the conclusions?

Reviewer #1: Yes

Reviewer #2: Yes

2. Has the statistical analysis been performed appropriately and rigorously? 

Reviewer #1: Yes

Reviewer #2: No

3. Have the authors made all data underlying the findings in their manuscript fully available?

Reviewer #1: Yes

Reviewer #2: No

4. Is the manuscript presented in an intelligible fashion and written in standard English?

Reviewer #1: Yes

Reviewer #2: Yes

5. Review Comments to the Author

Reviewer #1: This study wants to explore the value of the costimulatory molecule CD86 in the early diagnosis of sepsis in the emergency department and explore the value of HLA in the prognosis of patients with sepsis. The authors pointed out that CD86 is an independent predictor of the occurrence of sepsis in patients with infection in the emergency department and has a certain value in the early clinical diagnosis of sepsis in such patients.

The Manuscript needs some minor revisions:

-Improve the English form.

-It is unclear how control patients were enrolled. Do they always come from the emergency room? are they patients with fever? The authors should clarify the description of this group.

- the tables and figures are clear except for figure 2, the symbols are not clear and are small, also the figure should be divided into A and B.

Reviewer #2: The authors have written a nice manuscript which is a prospective study.

1. Variables with P<0.15 in the univariable regression analysis were selected for multivariable logistic regression analysis (MVA). Why not only variables with significant value p<0.05 were taken for MVA

2. Why were blood samples collected after 1st dose of antibiotics, and not at baseline?

3. How can variables with such low odds ratios [PLT (OR=0.99, 95%CI: 0.986-0.997, P=0.001), IL-10 (OR=1.01)] be statistically significant? What is the outcome of odds ratio for these variables?

4. In Methods - Please write clearly what was the primary end-point of the study. Writing to explore the value of CD86 and HLA-DR in sepsis, is very vague

5. How many patients who were initially classified as "non-sepsis" later turned into "sepsis" during the course of hospital stay? How were these patients accounted for?

6. PLOS authors have the option to publish the peer review history of their article (what does this mean?). If published, this will include your full peer review and any attached files.

Reviewer #1: No

Reviewer #2: **Yes: **Sumeet Mirgh

---

## [Author Response · Author response to Decision Letter 0]

16 Oct 2023

Title: Predictive value of CD86 for the occurrence of sepsis (Sepsis-3) in patients with infection

Journal: PLoS ONE

Response to Reviewers’ comments

Dear Editor, 

 We thank you for your careful consideration of our manuscript. We appreciate your response and overall positive initial feedback and made modifications to improve the manuscript. After carefully reviewing the comments made by the Reviewers, we have reorganized the literature data, modified the manuscript to improve the presentation of our results and their discussion, therefore providing a complete context for the research that may be of interest to your readers.

 We hope that you will find the revised paper suitable for publication, and we look forward to contributing to your journal. Please do not hesitate to contact us with other questions or concerns regarding the manuscript.

Best regards,

Keji Zhang

Reviewer #1 

The Manuscript needs some minor revisions:

-Improve the English form.

Response: We thank the Reviewer. The manuscript was proofread.

-It is unclear how control patients were enrolled. Do they always come from the emergency room? are they patients with fever? The authors should clarify the description of this group.

Response: We thank the Reviewer for the comment. In this study, the patients who visited the emergency room were enrolled if they were >18 years of age and had a confirmed infection or were with a high suspicion of infection with clear clinical symptoms. The patients who were highly suspected of having an infection with clear clinical symptoms were also required to meet at least two of the following four diagnostic criteria for systemic inflammatory response syndrome [1]: 1) body temperature of >38℃ or <36℃, 2) heart rate of >90 beats/min, 3) respiratory rate of >20 breaths/min, or partial pressure of carbon dioxide of <32 mmHg, and 4) peripheral white blood cell count of >12,000/mm3 or <4000/mm3.

Then, all patients with confirmed infection or highly suspected of infection were divided into the sepsis and the non-sepsis groups according to the Sepsis-3 standard [2]. Finally, the patients with sepsis were divided into the sepsis-survival and sepsis-death groups according to whether the patient died or not. Therefore, there were no healthy controls, and all patients had infection (or were highly suspected of). It was clarified in the manuscript.

- the tables and figures are clear except for figure 2, the symbols are not clear and are small, also the figure should be divided into A and B.

 Response: We thank the Reviewer for the comment. We agree that the group symbols were small and kind of lost in the line. Figure 2A represents the CD86%, HLA, and CD86 mRNA, while Figure 2B presents the flow cytometry outputs of CD86 among the three groups. It was revised accordingly.

Reviewer #2

The authors have written a nice manuscript which is a prospective study.

 Response: We thank the Reviewer for his/her appreciation of our work.

1. Variables with P<0.15 in the univariable regression analysis were selected for multivariable logistic regression analysis (MVA). Why not only variables with significant value p<0.05 were taken for MVA

Response: We thank the Reviewer for the comment. Due to the effect of other covariates, some characteristics not significant in the univariable regression analyses may be significant in the multivariable regression. Therefore, we selected variables with P<0.15 in the univariable regression analysis. In addition, the sample size was relatively small, particularly the sepsis-death groups, and using a higher P-value decreases the risk of missing a variable that would be significant in the multivariable analysis. P-values thresholds as high as 0.25 have been reported [3-5]. Still, we agree that using a more stringent threshold could change the variables included in the multivariable models. The results will have to be confirmed in larger multicenter studies. It was added in the Discussion.

2. Why were blood samples collected after 1st dose of antibiotics, and not at baseline?

Response: We are deeply sorry for the error that occurred during manuscript preparation and translation. The blood samples were taken after admission and before the first dose of antibiotics. It was corrected.

3. How can variables with such low odds ratios [PLT (OR=0.99, 95%CI: 0.986-0.997, P=0.001), IL-10 (OR=1.01)] be statistically significant? What is the outcome of odds ratio for these variables?

Response: We thank the Reviewer for the comment. It is because the determination of the odds ratio is based on the change of one unit of PLT, for example. Therefore, for each change of 1×109/L, the odds of sepsis decrease by 1%. Since the PLT levels are in the range of 91-316 ×109/L, a change of 1×109/L (which is small relative to the platelet levels) will lead to a small change in odds. It is a common observation with continuous variables in logistic regression models. For IL-10, the range was large, which could explain the results. In addition, in a multivariable analysis, each variable is adjusted for all the others [6-8].

4. In Methods - Please write clearly what was the primary endpoint of the study. Writing to explore the value of CD86 and HLA-DR in sepsis, is very vague

Response: We agree with the Reviewer. The primary study endpoint was the factors associated with a diagnosis of sepsis. The secondary study endpoint was the factors associated with mortality among patients with sepsis. It was clarified in the manuscript.

5. How many patients who were initially classified as “non-sepsis” later turned into “sepsis” during the course of hospital stay? How were these patients accounted for?

 Response: The Reviewer raises a good point. Twenty-three patients changed from non-sepsis to sepsis during hospitalization. They were grouped in the sepsis group after meeting the diagnostic criteria for sepsis. It was clarified in the Methods and Results.

Journal Requirements

1. Please ensure that your manuscript meets PLOS ONE’s style requirements, including those for file naming. The PLOS ONE style templates can be found at 

 Response: We revised the manuscript accordingly.

 Response: It was verified.

This work was supported by the Science and Technology Support Project for Medical Guidance (Chinese and Western Medicine) of Shanghai Municipal and the Science and Technology Commission in 2018 [No.18411967000]. The sponser had no role in study design, data collection and analysis, decision to publish, or preparation of the manuscript.

 Response: We are a little confused. The Journal asks here to add the funding information in the manuscript, but in the request just below, the Journal asks to remove all funding information from the manuscript. For now, we followed the instructions provided in point #1 and removed all funding statements from the manuscript.

 Response: We follow the instructions provided in point #1 and removed all funding statements from the manuscript. The funders had no role in study design, data collection and analysis, decision to publish, or preparation of the manuscript.

 Response: We are now providing the ORCID iD for the corresponding author.

 Response: It was revised.

 Response: We do not provide any supporting information.

 

References

1. Chakraborty RK, Burns B. Systemic Inflammatory Response Syndrome. StatPearls. Treasure Island (FL)2022.

2. Sansom DM, Manzotti CN, Zheng Y. What’s the difference between CD80 and CD86? Trends Immunol. 2003;24(6):314-9. Epub 2003/06/18. doi: 10.1016/s1471-4906(03)00111-x. PubMed PMID: 12810107.

3. Bursac Z, Gauss CH, Williams DK, Hosmer DW. Purposeful selection of variables in logistic regression. Source Code Biol Med. 2008;3:17. Epub 2008/12/18. doi: 10.1186/1751-0473-3-17. PubMed PMID: 19087314; PubMed Central PMCID: PMCPMC2633005.

4. Sauerbrei W, Perperoglou A, Schmid M, Abrahamowicz M, Becher H, Binder H, et al. State of the art in selection of variables and functional forms in multivariable analysis-outstanding issues. Diagn Progn Res. 2020;4:3. Epub 2020/04/09. doi: 10.1186/s41512-020-00074-3. PubMed PMID: 32266321; PubMed Central PMCID: PMCPMC7114804.

5. Chowdhury MZI, Turin TC. Variable selection strategies and its importance in clinical prediction modelling. Fam Med Community Health. 2020;8(1):e000262. Epub 2020/03/10. doi: 10.1136/fmch-2019-000262. PubMed PMID: 32148735; PubMed Central PMCID: PMCPMC7032893.

6. Ranganathan P, Pramesh CS, Aggarwal R. Common pitfalls in statistical analysis: Logistic regression. Perspect Clin Res. 2017;8(3):148-51. Epub 2017/08/23. doi: 10.4103/picr.PICR_87_17. PubMed PMID: 28828311; PubMed Central PMCID: PMCPMC5543767.

7. Roalfe AK, Holder RL, Wilson S. Standardisation of rates using logistic regression: a comparison with the direct method. BMC Health Serv Res. 2008;8:275. Epub 2008/12/31. doi: 10.1186/1472-6963-8-275. PubMed PMID: 19113996; PubMed Central PMCID: PMCPMC2661894.

8. Stoltzfus JC. Logistic regression: a brief primer. Acad Emerg Med. 2011;18(10):1099-104. Epub 2011/10/15. doi: 10.1111/j.1553-2712.2011.01185.x. PubMed PMID: 21996075.

---

## [Decision Letter · Decision Letter 1]

6 Feb 2024

PONE-D-23-11681R1Predictive value of CD86 for the occurrence of sepsis (Sepsis-3) in patients with infectionPLOS ONE

Dear Dr. Zhang,

Thank you for submitting your manuscript to PLOS ONE. After careful consideration, we feel that it has merit but does not fully meet PLOS ONE’s publication criteria as it currently stands. Therefore, we invite you to submit a revised version of the manuscript that addresses the points raised during the review process.

We look forward to receiving your revised manuscript.

Kind regards,

Chiara Lazzeri

Academic Editor

PLOS ONE

Journal Requirements:

Reviewers' comments:

Reviewer's Responses to Questions

**Comments to the Author**

1. If the authors have adequately addressed your comments raised in a previous round of review and you feel that this manuscript is now acceptable for publication, you may indicate that here to bypass the “Comments to the Author” section, enter your conflict of interest statement in the “Confidential to Editor” section, and submit your "Accept" recommendation.

Reviewer #3: (No Response)

2. Is the manuscript technically sound, and do the data support the conclusions?

Reviewer #3: Yes

3. Has the statistical analysis been performed appropriately and rigorously? 

Reviewer #3: Yes

4. Have the authors made all data underlying the findings in their manuscript fully available?

Reviewer #3: No

5. Is the manuscript presented in an intelligible fashion and written in standard English?

Reviewer #3: Yes

6. Review Comments to the Author

Reviewer #3: The authors present a well performed observational study and the manuscript is well organized and written. The control group with infection and signs of systemic inflammation is well chosen and reflects the clinical question sepsis biomarkers are needed for.

To my understanding the data availability does not fulfill PLOS requirements. The data should either be available from a repository or you should explain why not. Apart from that the manuscript is suitable or publication.

Apart from that I have only minor remarks that might be considered by the author but are not relevant for the deission to publish. The discussion is a little bit lengthy reviewing a lot of molecular biology mainly of interest to a small section of the potentiall readership. Maybe citing a good review cold shorten it, but as mentioned this might just be my personell preference.

In table one I am not sure if you relly should test for significant differences in septic shock, ventialition or SOFA scores. All these variables ae hallmarks of organ failure and therefore the fulfillment of sepsis-3 criteria and the differences between the sepsis and non-sepsis group are therefore not random by design.

7. PLOS authors have the option to publish the peer review history of their article (what does this mean?). If published, this will include your full peer review and any attached files.

Reviewer #3: No

---

## [Author Response · Author response to Decision Letter 1]

19 Mar 2024

Title: Predictive value of CD86 for the occurrence of sepsis (Sepsis-3) in patients with infection

Journal: PLOS ONE

Response to Reviewers' comments

Dear Editor, 

 We thank you for your careful consideration of our manuscript. We appreciate your response and overall positive initial feedback and made modifications to improve the manuscript. After carefully reviewing the comments made by the Reviewers, we have reorganized the literature data, modified the manuscript to improve the presentation of our results and their discussion, therefore providing a complete context for the research that may be of interest to your readers.

 We hope that you will find the revised paper suitable for publication, and we look forward to contributing to your journal. Please do not hesitate to contact us with other questions or concerns regarding the manuscript.

Best regards,

Keji Zhang

Academic Editor

Journal Requirements:

Response: We thank the Academic Editor. We verified our references. We did not cite any references flagged as retracted on PubMed.

Reviewer #3

1. If the authors have adequately addressed your comments raised in a previous round of review and you feel that this manuscript is now acceptable for publication, you may indicate that here to bypass the “Comments to the Author” section, enter your conflict of interest statement in the “Confidential to Editor” section, and submit your "Accept" recommendation.

Reviewer #3: (No Response)

2. Is the manuscript technically sound, and do the data support the conclusions?

Reviewer #3: Yes

3. Has the statistical analysis been performed appropriately and rigorously?

Reviewer #3: Yes

4. Have the authors made all data underlying the findings in their manuscript fully available?

Reviewer #3: No

 Response: We thank the Reviewer for the comment. As per local regulations in our hospital, we are not allowed to upload our data on public registries for now, but the data can be made available to researchers upon reasonable request. As a result, we are unable to provide the requested deposition at this time. We understand the importance of these processes and will continue to explore possibilities for compliance with such requirements in the future.

5. Is the manuscript presented in an intelligible fashion and written in standard English?

Reviewer #3: Yes

6. Review Comments to the Author

Reviewer #3: The authors present a well performed observational study and the manuscript is well organized and written. The control group with infection and signs of systemic inflammation is well chosen and reflects the clinical question sepsis biomarkers are needed for.

To my understanding the data availability does not fulfill PLOS requirements. The data should either be available from a repository or you should explain why not. Apart from that the manuscript is suitable or publication.

 Response: We thank the Reviewer for the comment. As per local regulations in our hospital, we are not allowed to upload our data on public registries, but the data can be made available to researchers upon reasonable request. As a result, we are unable to provide the requested deposition at this time. We understand the importance of these processes and will continue to explore possibilities for compliance with such requirements in the future.

Apart from that I have only minor remarks that might be considered by the author but are not relevant for the deission to publish. The discussion is a little bit lengthy reviewing a lot of molecular biology mainly of interest to a small section of the potentiall readership. Maybe citing a good review cold shorten it, but as mentioned this might just be my personell preference.

 Response: We thank the Reviewer for the comment. We agree with the Reviewer that this study did not examine the exact molecular biology and that the discussion, even if explanatory, might be superfluous to many of the readership. Therefore, as suggested, we cited good reviews, but kept the discussion considering that it explains our results. Hence, the interested readership will be able to seek more information.

In table one I am not sure if you relly should test for significant differences in septic shock, ventialition or SOFA scores. All these variables ae hallmarks of organ failure and therefore the fulfillment of sepsis-3 criteria and the differences between the sepsis and non-sepsis group are therefore not random by design.

 Response: We understand the Reviewer’s comment. We simply wanted to show the characteristics of the patients.

---

## [Editor Report · Decision Letter 2]

27 Mar 2024

Predictive value of CD86 for the occurrence of sepsis (Sepsis-3) in patients with infection

PONE-D-23-11681R2

Dear Dr. Zhang,

We’re pleased to inform you that your manuscript has been judged scientifically suitable for publication and will be formally accepted for publication once it meets all outstanding technical requirements.

Kind regards,

Chiara Lazzeri

Academic Editor

PLOS ONE
---

## [Editor Report · Acceptance letter]

1 Apr 2024

PONE-D-23-11681R2 

PLOS ONE

Dear Dr. Zhang, 

I'm pleased to inform you that your manuscript has been deemed suitable for publication in PLOS ONE. Congratulations! Your manuscript is now being handed over to our production team.

Kind regards, 

on behalf of

Dr. Chiara Lazzeri 

Academic Editor

PLOS ONE